# CircuitNet 2.0: An Advanced Dataset for Promoting Machine Learning Innovations in Realistic Chip Design Environment

**Xun Jiang**[1], **Zhuomin Chai**[1,2], **Yuxiang Zhao**[1], **Yibo Lin**[1]*, **Runsheng Wang**[1], **Ru Huang**[1]

[1]Peking University [2]Wuhan University

xunjiang@stu.pku.edu.cn, zhuominchai@whu.edu.cn, yuxiangzhao@stu.pku.edu.cn
{yibolin,r.wang,ruhuang}@pku.edu.cn

## Abstract

Integrated circuits or chips are key to enable computing in modern industry. Designing a chip relies on human experts to produce chip data through professional electronic design automation (EDA) software and complicated procedures. Nowadays, prompted by the wide variety of machine learning (ML) datasets, we have witnessed great advancement of ML algorithms in computer vision, natural language processing, and other fields. However, in chip design, high human workload and data sensitivity cause the lack of public datasets, which hinders the progress of ML development for EDA. To this end, we introduce an advanced large-scale dataset, CircuitNet 2.0 [1], which targets promoting ML innovations in a realistic chip design environment. In order to approach the realistic chip design space, we collect more than 10,000 samples with a variety of chip designs (e.g., CPU, GPU, and AI Chip). All the designs are conducted through complete commercial design flows in a widely-used technology node, 14nm FinFET. We collect comprehensive data, including routability, timing, and power, from the design flow to support versatile ML tasks in EDA. Besides, we also introduce some realistic ML tasks with CircuitNet 2.0 to verify the potential for boosting innovations.

## 1 Introduction

Chip design is a sophisticated and complex process, which can be roughly decomposed into three procedures, including abstract chip design, EDA flow, and physical manufacture, as shown in Figure 1. More detailed descriptions of chip design flow are illustrated in Figure A.1. The goal of EDA flow is to help hardware designers produce manufacturable chips with better performance, power, and area (PPA). These design targets are hard to meet simultaneously considering the growing scale of hardware designs. In practice, EDA flow for digital circuits can be divided into logical synthesis, physical design, analysis, and verification. The logical synthesis converts hardware description codes (e.g., Verilog codes) to a synthesized netlist, which defines the topological relationships (i.e., nets) between basic functional elements (i.e., logic cells and pre-designed macros). The physical design flow aims at generating the final manufacturable chip description from the synthesized netlist through a series of processing stages, e.g., floorplan, powerplan, placement, clock tree synthesis (CTS), and routing. Each stage tries to tackle a complex optimization problem with different objectives and constraints. At the end of each stage, timing and power analysis can be conducted to evaluate the quality of designs. To catch up with the emerging chips (Liu et al., 2024) and design methodologies (He et al., 2023; Gao et al., 2023), some tools fuse consecutive design stages to enable larger solution space and achieve better overall quality. However, realistic chip design environment always contains various design types, design scales, manufacturing technologies, and desired targets, where the quality, efficiency, robustness, and reliability of EDA tools encounter serious challenges. In cutting-edge EDA tools, optimization at one stage tries to leverage the predictive results at downstream stages by ML to improve the eventual quality of results (QoR) (Rapp et al., 2022) and efficiency.

---

*corresponding author
[1]https://circuitnet.github.io/

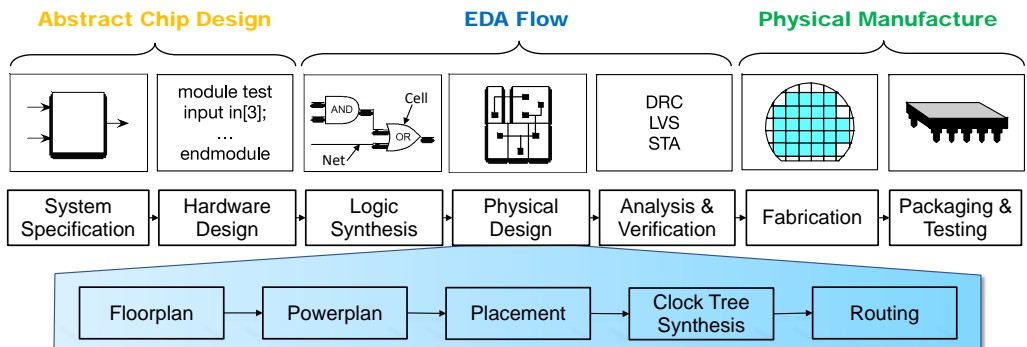

Figure 1: Typical chip design flow for digital circuits.

Although ML has been introduced into EDA, there are very few public datasets available to implement and evaluate ML algorithms, compared with other fields like computer vision, natural language processing, speech recognition, and recommendation systems. The EDA community has been expecting a large-scale and effective dataset like ImageNet (Deng et al., 2009) in computer vision for ML tasks in EDA. However, an ML for EDA dataset needs to be generated by numerous runs of EDA flows with domain knowledge in chip design and each run can take weeks to months, making it extremely expensive for data generation. Furthermore, the Non-disclosure Agreement (NDA) in manufacturing technology and EDA tools also prohibits the release of the original data. As a result, most studies can only generate small internal datasets for validation of ML techniques, making benchmarking and reproducing results extremely challenging. This situation also places a high bar for new researchers and limits the research inside EDA community. Previous efforts like CircuitNet (Chai et al., 2022; 2023) try to alleviate such situation by releasing an open-source dataset containing over 10,000 samples and supports ML tasks in EDA, e.g., routability and IR-drop prediction (Ren & Hu, 2022). However, the design types, design scales, manufacturing technology, and supported ML tasks are still far from representing the realistic chip design environment.

In this work, we present CircuitNet 2.0, another advanced open-source ML for EDA dataset, to promote more ML innovations in the realistic chip design environment. The dataset contains data to support multi-model prediction tasks in advanced technology nodes, like timing, routability, and IR-drop, covering more design objectives compared with CircuitNet. CircuitNet 2.0 goes beyond the original CircuitNet in three perspectives: 1) Providing data for million-gate designs like CPU, GPU, and AI chips to increase circuit types and scales; 2) Generating data with advanced 14nm FinFET technology node to increase the manufacturing and modeling complexity; 3) Adding an additional timing prediction task for circuit performance to increase task diversity and practicality. The major differences are summarized in Figure 2(a). Therefore, in this paper, we introduce how to formulate the prediction tasks with multi-modal data, and also provide commonly-used features from recent studies to help new researchers get started quickly. Besides, we also introduce some realistic ML tasks in this paper, e.g., ML tasks based on imbalanced samples, transfer learning across different technology nodes (e.g., process design kits (PDKs)) [2] and designs.

## 2 RELATED WORK

### 2.1 DATASETS AND BENCHMARKS FOR CHIP DESIGN

Datasets and benchmarks are critical for the research community. The well-known conferences, e.g. ISPD and ICCAD, host contests about EDA problems, which provide benchmarks with pre-processed data for the researchers. However, most of these benchmarks only contain isolated data used for specific tasks, usually placement or routing, which cannot be directly used for ML tasks and constraints further cross-stage innovation. OPDB (Tziantzioulis et al., 2021) is a scalable, modular, heterogeneous, and extensible design benchmark, which allows users to generate hardware designs conveniently. The EPFL combinational benchmark suite (Amarú et al., 2015) provides data to challenge modern logic optimization tools. These original hardware designs enable flexibility for the

---

[2]A process design kit (PDK) is a set of files associated with specific technology to model a fabrication process for the integrated circuit design.

users to utilize, but their application scopes are limited to a specific task due to lack of intermediate data dumped from the EDA flow. BeGAN (Chhabria et al., 2021a) proposed GAN and transfer learning-based methodology for synthetic Power Delivery Network (PDN) benchmark generation. The data from BeGAN only support IR-drop prediction and its quality may not be comparable to that generated by commercial EDA tools. Kim et al. (Kim et al., 2022) proposed an artificial netlist generation flow based on open-source EDA tools OpenRoad (Ajayi et al., 2019). OpenABC-D (Chowdhury et al., 2021) is a dataset proposed for ML applications in logic synthesis without running the physical design flow. Compared with previous efforts, our dataset provides the cross-stage multi-modal data generated by commercial EDA flow and tools with a convenient data format for a variety of ML tasks.

## 2.2 PRACTICAL PREDICTION TASKS

The basic problem encountered in EDA is a series of large-scale NP-hard combinatorial optimization problems. The EDA tools nowadays need to deal with billions of transistors, properly place them on the chip and connect them to achieve the best performance, which is called the physical design of chips, and heuristics are widely used to obtain an optimized result. Although many efforts have tried to directly solve the combinatorial optimization problem using ML techniques, like macro placement with reinforcement learning (Mirhoseini et al.), recent results show that such an approach still cannot compete with traditional methods or well-tuned heuristics (Cheng et al.). Compared to directly solving the problem, using ML models to predict tool outcomes at early design stages and guide heuristic optimization has become a popular approach in EDA, offering faster convergence and reduced pessimism. This dataset focuses on optimizing three most important design objectives, i.e., routability, IR-drop, and timing. Routability measures how easily a chip design can be routed; typical metrics include routing congestion and distributions of design rule violations (DRV). IR-drop refers to the voltage drop on power supply signals due to the current and resistance on power supply wires. Timing measures the performance of chips, including the delay of cells and nets.

As a chip comprises both geometric information like the locations of cells, and topological information like the connectivity of cells (i.e., nets), we can leverage such multi-modal information to build ML models. Existing approaches can be roughly divided into image-based modeling and graph-based modeling.

**Image-based Modeling**: In physical design, many tasks can be formulated as image-to-image translation tasks, as the geometric information on the chip can be represented as an image. This includes predictions related to routing congestion, DRV hotspots, and IR-drop, which are defined as follows. Congestion is defined as the overflow of routing demand over routing resources. Its prediction can be formulated as an image-to-image translation task with generative models in early studies (Alawieh et al., 2020; Chen et al., 2020b; Liu et al., 2021; Cheng et al., 2022). DRV hotspots are positions with undesirable designs that will become defects in manufacturing. Its prediction is also based on images as an analogy to image recognition (Chan et al., 2017; Xie et al., 2018; Islam & Shahjalal, 2019; Yu et al., 2019; Tabrizi et al., 2019; Liang et al., 2020; Hung et al., 2020; Baek et al., 2022). IR-drop is the deviation of supply power due to large local current demand. Its prediction can be performed on images, but the time domain is also introduced to consider different operating conditions, making the input similar to images with time sequence, i.e., video. Simple 2D and 3D CNN models are effective in this task (Xie et al., 2020; Chhabria et al., 2021b).

**Graph-based Modeling**: Modeling with graphs has been introduced into tasks that largely rely on topological information and require higher granularity, such as congestion prediction (Wang et al., 2022; Yang et al., 2022) and timing prediction (Xie et al., 2021; Guo et al., 2022; Lopera & Ecker, 2022; Ye et al., 2023; Chhabria et al., 2022), to preserve the details and topological information lost in forming images through tiling the original chip. For example, net delay prediction has to leverage graph structure to model the correlation between adjacent cells and nets. This approach requires more effort in constructing a graph, so it is not as widely adopted as image-based modeling, but recent works show better accuracy can be achieved with this approach (Wang et al., 2022; Yang et al., 2022).

## 2.3 REALISTIC CHIP DESIGN ENVIRONMENT

In the realistic chip design environment, most designs are universal chips (e.g., CPU) rather than specialized chips (e.g., GPU and AI Chip), which forms a naturally imbalanced distribution of chip

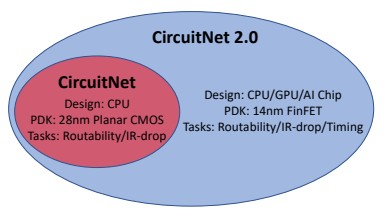

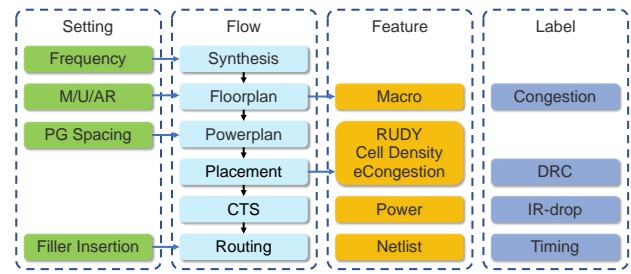

(a) The difference between CircuitNet 2.0 and CircuitNet.

(b) Setting, flow, features, and labels for various tasks are shown in the design flow. These items are explained in Appendix A.3.

Figure 2: The overview of CircuitNet 2.0 and the difference compared with CircuitNet.

Table 1: Statistics of designs in CircuitNet 2.0.

| Design | #Cells | #Nets | #Macros | #Pins | #IOs | #Samples |
|---|---|---|---|---|---|---|
| RISCY | 46,184 | 47,233 | 3 | 180,069 | 563 | 3,456 |
| RISCY-FPU | 65,464 | 66,903 | 3 | 252,390 | 563 | 3,456 |
| zero-riscy | 35,969 | 36,225 | 3 | 138,569 | 563 | 3,456 |
| OpenC910-1 | 754,981 | 766,436 | 32 | 3,062,504 | 1,341 | 96 |
| Vortex-large | 1,018,221 | 1,107,255 | 376 | 3,731,139 | 1,242 | 74 |
| Vortex-small | 113,961 | 124,058 | 43 | 433,449 | 1,234 | 96 |
| NVDLA-large | 1,478,865 | 1,637,556 | 80 | 5,705,108 | 1,734 | 68 |
| NVDLA-small | 270,072 | 285,465 | 108 | 1,039,571 | 538 | 89 |

[†] The smallest design zero-riscy takes about **2 hours** to generate one data sample, while the largest design NVDLA-large takes nearly **1 week** to generate one data sample.

data. Furthermore, designing a large-scale chip (e.g., high-performance GPU and AI Chip) will take over two orders of magnitude longer than designing a small-scale chip (e.g., micro-controller CPU). These factors make the chip data imbalance a problem that researchers have to face. Meanwhile, with the development of technologies, transferring the knowledge from previous generation chips to improve the efficiency for the development of new generalization chips is widely adopted in the realistic chip design environment. The literature (Xie et al., 2018; Gai et al., 2022; Lin et al., 2019) has investigated the benefits of applying transfer learning in chip design.

## 3 DATASET

### 3.1 OVERVIEW OF CIRCUITNET 2.0

CircuitNet 2.0 is generated by commercial design tools with advanced technology node, 14nm FinFET, and various designs, whose types and scales are widely different. The difference between CircuitNet 2.0 and CircuitNet are shown in Figure 2a. CircuitNet 2.0 includes richer design types ( e.g., CPU, GPU, and AI Chip) and advanced PDK (e.g., 14nm FinFET) and supports additional timing prediction tasks, while only CPU designs and 28nm planar CMOS PDK are employed in the original CircuitNet. In practice, the samples within CircuitNet 2.0 and CircuitNet are orthogonal, which could be used together to further enrich the sample diversity for ML tasks. We demonstrate the generation flow of CircuitNet 2.0 in Figure 2b, including setting, design flow, feature, and label for ML tasks. The setting column on the left of the figure indicates the options of setting to generate different samples. The flow column consists of synthesis, floorplan, powerplan, placement, CTS, and routing. In practice, routing is executed in two phases, including global routing (GR) and detailed routing (DR). We leverage Design Compiler (Synopsys, 2021.06) for synthesis and Innovus (Cadence, 2020.10/2021.11) for the other physical design stages.

Features and labels are extracted from the flow as shown in the right two columns in Figure 2b. The geometric information of macros and cells is mainly extracted from floorplan and placement stages. Some hand-crafted features (e.g., RUDY), are further generated by raw physical information with a

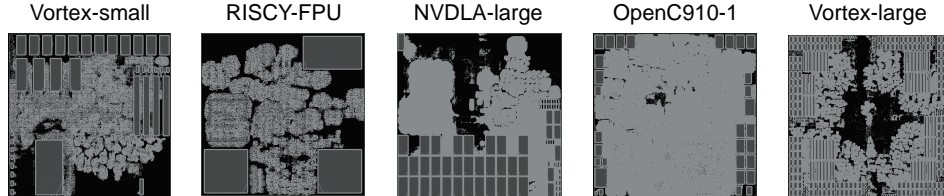

| Vortex-small | RISCY-FPU | NVDLA-large | OpenC910-1 | Vortex-large |
|---|---|---|---|---|

Figure 3: Visualization of the placement of hardware designs.

Table 2: Features and labels for various ML tasks based on CircuitNet 2.0.

| Prediction Task | Features | | Label | |
|---|---|---|---|---|
| | Name | Type | Name | Type |
| Routability | Macro Region | Image | Congestion | Image |
| | RUDY | Image | | |
| | Instance Placement | Graph | | |
| | Macro Region | Image | DRV | Image |
| | Cell Density | Image | | |
| | RUDY | Image | | |
| | Pin Configuration | Image | | |
| | eCongestion | Image | | |
| | Congestion | Image | | |
| IR-drop | Overall Power | Image | IR-drop | Image |
| | Temporal Power | 3D Array | | |
| Timing | Netlist | Graph | Net delay | Graph |
| | Pin Position | Graph | | |

[†] Image in the table is 2D array by default.

small time cost. Power and timing data can be extracted at all stages of the flow but with different precisions. The topological information of netlists can also be extracted at all stages to capture the accurate relationship between macros and cells. Besides, congestion and DRV are extracted at the routing stage to reflect the final routability.

We list the statistics of the designs in Table 1. The total number of designs in CircuitNet 2.0 is 10,791. The scales of designs are distributed over a large range, e.g., the cell number of the largest design NVDLA-large (1,478,865) is almost 40× larger than the smallest design zero-riscy (35,969), while the maximal cell number of designs in CircuitNet is below 70,000. The types in CircuitNet 2.0 include CPU, GPU, and AI Chip, which are more diverse than CPU in CircuitNet. Considering the quite different sample generation time attached in Table 1, the number of large design samples is overall enough within restricted time budgets.

## 3.2 NETLIST, LAYOUT, AND OTHER FEATURES

Netlist and layout are two pillars to represent the chip in the physical design stage in the format of topological and geometric data. The raw data of netlist and layout can be viewed as features for ML tasks. Besides, other features, including hand-crafted or tool-analyzed data, are also introduced in the ML tasks. These data make the ML tasks in chip design multi-modal naturally.

Netlist is used to represent the logical relationship of synthesized circuits, which are composed of nets, cells, macros and IOs. Each cell has pre-defined pins by the PDK. By these elements, a netlist can be inherently defined as a hypergraph $\mathcal{H}(\mathcal{V}, \mathcal{E})$. The cells, macros, and IOs correspond to vertices $\mathcal{V}$ in the hypergraph $\mathcal{H}$. The nets correspond to the hyperedges $\mathcal{E}$, where each hyperedge $e$ is a subset of vertices of $\mathcal{V}$. In the hypergraph $\mathcal{H}$, the degree $d$ of a macro vertex is usually much larger than that of a cell vertex or an IO vertice. Besides, another hypergraph $\bar{\mathcal{H}}(\bar{\mathcal{V}}, \bar{\mathcal{E}})$ can also be defined by the elements in netlists. The pins and IOs are defined as vertices $\bar{\mathcal{V}}$ in the hypergraph $\bar{\mathcal{H}}$. IOs can

also be viewed as vertices because there is normally one pin in each IO. By this definition, the cells and nets correspond to the hyperedges $\bar{\mathcal{E}}$, as they mean the connections of several pins in netlists. In Table 1, the numbers of each component in different designs are listed to show the scale of related optimization or prediction problems. We list the average numbers of cells, nets, macros, pins, and primary IO pins for each design, as well as the number of samples in the dataset.

Layout is used to depict the size, shape, and position of the chip and the logic elements within it. We demonstrate the visualizations of placement belonging to five different designs in Figure 3. `RISCY-FPU` and `OpenC910-1` are CPU designs. `Vortex-small` and `Vortex-large` are GPU designs. `NVDLA-large` is an AI Chip. In the placement view, these clumps of gray matter are placed cells, whose sizes are much smaller than macros. The adjacent black regions are vacated by the utilization of cells. These chips have different sizes but are resized to the same figure sizes for easier visualization.

Other features extracted from CircuitNet 2.0, e.g., RUDY, congestion, power, timing, etc., are listed in Table 2. Due to the most features are extracted from various intermediate stages of the flow, some of them can be viewed as labels for the specific prediction task. The features and labels are illustrated by name, type, and related ML prediction tasks.

# 4 REALISTIC MACHINE LEARNING TASKS

In CircuitNet 2.0, we generate samples with advanced 14nm FinFET PDK and commercial tools to approach the realistic chip design quality. Meanwhile, multiple hardware, such as CPU, GPU, and AI Chip, are included to increase the diversity of our dataset, whose composition of different types of design gets closer to the realistic chip design environment than CircuitNet. With the help of CircuitNet 2.0, we introduce two realistic ML tasks in chip design, machine learning based on imbalanced data and transfer learning between PDK and designs.

## 4.1 LEARNING ON MULTI-MODAL DATA

CircuitNet 2.0 supports multiple multi-modal ML tasks in chip design, including routability prediction, IR-drop prediction, and timing prediction shown in Table 2. In this section, we take timing prediction task as an example. More detailed experimental results about other tasks based on CircuitNet 2.0 are provided in Appendix B.

### 4.1.1 TASK DESCRIPTION

Timing is a critical metric to measure the correctness of the propagation of logical functions through the netlists, which also reflects the margin and degree of violation of the current designs under specific time budgets. Therefore, the procedure of timing analysis is to propagate delays between the pin vertices in the hypergraph $\mathcal{H}$ defined in Section 3. In the design flow, timing analysis can be performed after almost all stages, e.g., placement, CTS, and routing, but at different accuracy. The later stage, the more accurate analysis can be performed. The goal of timing prediction is to accurately predict timing at early design stages without running the entire EDA flow.

We take the net delay prediction as an example, where the labels are edge features representing the propagation delays between pins of netlists. This task contains multi-modal data, i.e., topological connectivities and geometric information. One way to formulate this problem is as a graph-based prediction problem. The topological connectivities, positions in the layout, electronic properties, and geometric shapes of pins can be encoded as the input node features. Some geometric properties of nets can also be encoded as edge features in the graph. Users can either build the graphs from the provided features or use our prebuild graphs.

### 4.1.2 EXPERIMENTS

In this section, we present the prediction of net delays after routing by the outcomes of placement. Due to the net delay reflecting the propagation time of signals along connection wires, the actual timing graph $\mathcal{G}$ is constructed as shown in Figure 4. The nodes of the timing graph $\mathcal{G}$ are pins and edges are connections along signal propagation directions. Pin positions are used as node features to predict net delay.

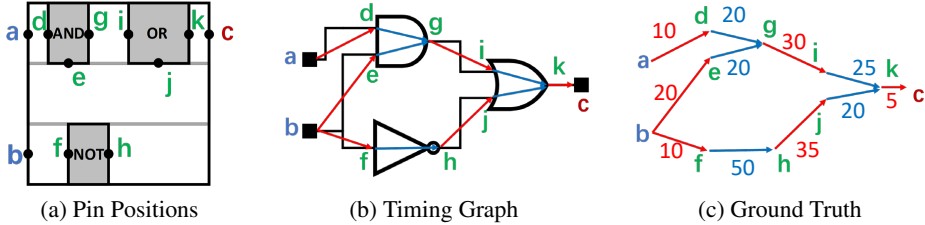

| (a) Pin Positions | (b) Timing Graph | (c) Ground Truth |

Figure 4: Predict net delay in timing graph at placement stage. Blue arrows are cell edges. The red arrows are net edges. The numbers on net edges are net delays to predict.

Table 3: Results of net delay prediction.

| Dataset | Model | $R^2$ score |
|---------|-------|-------------|
| CircuitNet 2.0 | MLP | $0.5223 \pm 0.017$ |
| CircuitNet 2.0 | GNN | $0.9262 \pm 0.005$ |

For CircuitNet 2.0, we use 80 samples from `RISCY`, `RISCY-FPU` and `Vortex` as the training set, while using 20 samples from `zero-riscy`, `NVDLA` and `OpenC910-1` as the test set. The samples are randomly selected and fixed for all experiments.

We implement two models for this task, a multilayer perceptron (MLP) and a GNN. The MLP has five hidden layers, each with 32 hidden neurons, and it simply takes the node features related to the net to fit the net delay. The GNN is modified from (Guo et al., 2022), leaving only the net embedding model to predict net delay. We use Adam optimizer with a learning rate of $5 \times 10^{-4}$ and mean-square error (MSE) Loss for all experiments.

**Evaluation metrics.** We adopt the $R^2$ score to evaluate the prediction results.

**Results.** The prediction results are shown in Table 3. We find the performance of MLP is lower than that of GNN, which indicates the graph structure can better capture the behavior of signal propagation and predict net delays.

## 4.2 LEARNING ON IMBALANCED DATA

The data imbalance problem refers to an unequal distribution of data, where the minority class only occupies a very small portion. From Table 1, the number of small designs, e.g. `RISCY`, `RISCY-FPU` and `zero-riscy`, take up a quite large portion of CircuitNet 2.0, which causes the dataset imbalanced. The source of this situation originates from the diverse costs of collecting data with different design sizes and sample types, which is common in the realistic chip design environment. In practice, the runtime costs of small designs, e.g. `RISCY`, `RISCY-FPU` and `zero-riscy`, are affordable for us (around 2 hours per sample), when we build a large-scale dataset. On the opposite, the samples of large designs, e.g. `OpenC910-1`, `Vortex-large`, and `NVDLA-large` are much fewer due to unaffordable runtime costs in generation (nearly 1 week per sample).. In this section, we analyze the data imbalance problem in detail from the perspectives of cross-design validation, t-SNE visualization, and oversampling study.

### 4.2.1 CROSS-DESIGN VALIDATION

We conduct cross-design validation in congestion prediction based on the FCN (Liu et al., 2021) on CircuitNet 2.0, in which we have three groups of samples, i.e., CPU, GPU, and AI Chip. We train on each group and test on the others using the same training setup. We use normalized root-mean-square error (NRMSE) and structural similarity index measure (SSIM) to evaluate the pixel-level accuracy and image similarity. The results are shown in Table 4. However, we interestingly observe that the model can still behave in a preferable generalization manner between different designs, which indicates the limited impact of data imbalance in the congestion prediction task.

Table 4: Results of cross-design validation in congestion prediction.

| | CPU as Train Set | | | GPU as Train Set | | | AI Chip as Train Set | |
|---|---|---|---|---|---|---|---|---|
| Test Set | NRMSE | SSIM | Test Set | NRMSE | SSIM | Test Set | NRMSE | SSIM |
| GPU | 0.0805 | 0.6816 | AI Chip | 0.0730 | 0.6735 | CPU | 0.0362 | 0.8520 |
| AI Chip | 0.0730 | 0.6735 | CPU | 0.0470 | 0.7939 | GPU | 0.0562 | 0.8102 |

Table 5: Performance of our proposed oversampling method on CircuitNet 2.0.

| | GPU as Train Set | | | | AI Chip as Train Set | | |
|---|---|---|---|---|---|---|---|
| Test Set | Oversampling | NRMSE | SSIM | Test Set | Oversampling | NRMSE | SSIM |
| CPU | No | 0.0470 | 0.7939 | CPU | No | 0.0362 | 0.8520 |
| AI Chip | | 0.1379 | 0.6903 | GPU | | 0.0562 | 0.8102 |
| CPU | Yes | **0.0411** | **0.8439** | CPU | Yes | **0.0368** | **0.8595** |
| AI Chip | | **0.0471** | **0.7989** | GPU | | 0.0530 | **0.8261** |

### 4.2.2 t-SNE Visualization.

We plot the t-distributed stochastic neighbor embeddings (t-SNE) on the input features (Macro Region, RUDY, and pin RUDY) of three groups of designs, as shown in Figure 5. We see the distributions of the three groups of designs are aligned to some extent, which is consistent with the generalization ability in Table 4.

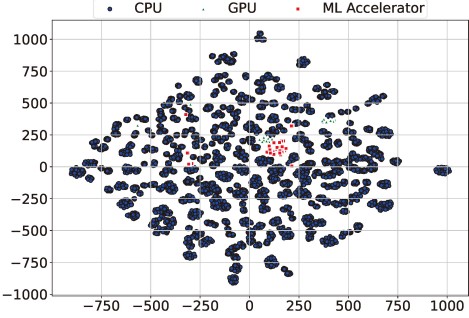

Figure 5: The t-SNE plot of the input features for congestion prediction in CircuitNet 2.0.

### 4.2.3 Oversampling Study

Furthermore, we also conducted an oversampling study to improve the generalization ability based on the imbalanced distribution. As shown in Table 5, the ablation study of oversampling is conducted when choosing GPUs or AI Chips as the training set. With the oversampling method, the performance of prediction is increased when the number of training samples is limited and fixed in a small range. That shows the data imbalance problem has the probability to be solved by oversampling a small amount of data.

As we all know, the data imbalance issue has always deeply plagued the field of EDA for a long time. We can alleviate this problem through a series of effective data sampling strategies (e.g., the abovementioned oversampling method) based on CircuitNet 2.0. More robust algorithms can be applied here, but this is not the focus of this work. The primary objective of this dataset is to collect data related to chip design as much as possible, thereby reducing the barriers to new researchers. As for the optimal manner of dataset usage, we leave this as an open problem for users to explore and propose better solutions.

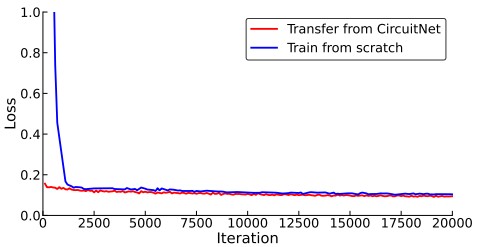

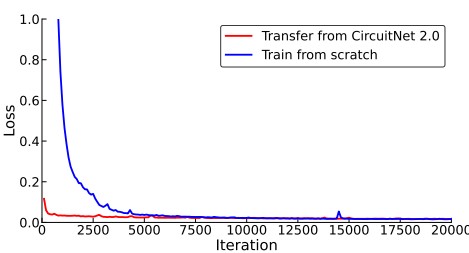

(a) Training curves of training from scratch v.s. transfer from CircuitNet Chai et al. (2023).

(b) Training curves of training from scratch v.s. transfer from CircuitNet 2.0 to CircuitNet-ISPD15 (Chai et al., 2023; Bustany et al., 2015).

Figure 6: Transfer learning results between different (a) technology nodes; (b) technology nodes and designs.

### 4.3 TRANSFER LEARNING BETWEEN TECHNOLOGY NODES AND DESIGNS

When applying ML techniques in the realistic chip design environment, the generalization and transferability between different PDKs and designs are useful in practice. The literature (Xie et al., 2018; Gai et al., 2022; Lin et al., 2019) has investigated the benefits of leveraging the pre-trained model from a larger dataset to improve the generalization effect on the smaller dataset. With the help of CircuitNet 2.0, we also conduct experiments to examine the possible generalization and transferability between different PDKs and designs. In the experiments, the learning rate of transfer learning is the same as learning from scratch.

We take the congestion prediction task as an example and use the FCN model (Liu et al., 2021) for transfer learning. First, we transfer the knowledge learned from CircuitNet to CircuitNet 2.0, which is from an older technology node to an advanced technology node. We show the training curves of "Training from scratch" and "Training from CircuitNet" on CircuitNet 2.0 in Figure 6a. With the pre-trained model, much faster convergence can be reached compared with training from scratch.

Second, we introduce the samples from ISPD15 contest benchmark (Bustany et al., 2015) in CircuitNet, denoted as CircuitNet-ISPD15 (Chai et al., 2023), to demonstrate the transfer learning between different technology nodes and designs simultaneously. CircuitNet-ISPD15 is developed from ISPD 2015 placement contest benchmarks Bustany et al. (2015) which is originally used for validating placement algorithms. The ISPD 2015 benchmark contains five pre-synthesized designs named "superblue" with about 1 million cells. There are 150 samples in CircuitNet-ISPD15 through the same data generation flow and PDK as CircuitNet. We show the curves of "Training from scratch" and "Training from CircuitNet 2.0" on CircuitNet-ISPD15 in Figure 6b. An interesting observation is that the loss of "Transfer from CircuitNet 2.0" drops quickly at the first 500 iterations, unlike the loss of "Transfer from CircuitNet" in Figure 6a that drops steadily during the entire training process. This result is reasonable, as the designs in CircuitNet-ISPD15 are different from those in CircuitNet 2.0 and thus the model needs to adapt to the "unseen" designs at the beginning iterations of training, while CircuitNet 2.0 and CircuitNet partly share similar CPU designs.

## 5 CONCLUSION

Our work provides an advanced dataset CircuitNet 2.0, implemented with 14nm FinFET technology node, for promoting ML innovations in the realistic chip design environment. The dataset contains over 10,000 samples, including CPU, GPU, and AI Chip from academia and industry. The dataset is generated based on commercial EDA tools and PDK, which shows high consistency with the realistic chip design environment. The features and labels in the dataset can be used in various prediction tasks. We also conduct multiple realistic machine learning tasks in chip design, i.e., learning on multi-modal data, imbalanced data, and transfer learning between technology nodes and designs. In the future, we will continue to enrich the diversity of the dataset and investigate variations of EDA tools and flows. We hope this work to promote the research for ML in realistic chip design and pave the way for AI for EDA researchers.

## ACKNOWLEDGMENTS

This work is supported in part by National Science and Technology Major Project (Grant No. 2021ZD0114702), the Natural Science Foundation of China (Grant No. T2293701 and No. 62034007), and the 111 project (B18001).

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

## A  COMPLEMENTARY DESCRIPTION OF THE DATASET

In the appendix, we give a more comprehensive description of our dataset.

### A.1  HARDWARE DESIGNS

We choose typical hardware, e.g., CPU, GPU, and AI Chip, to generate different samples. All these designs are from open-source projects, which can be easily achieved from the Internet. Here, we list the links to these projects and their licenses.

1. `PULPino` (CPU with Solderpad Hardware License):
   `https://github.com/pulp-platform/pulpino`.
2. `OpenC910` (CPU with Apache License 2.0):
   `https://github.com/T-head-Semi/openc910`
3. `Vortex` (GPU with BSD-3-Clause License):
   `https://github.com/vortexgpgpu/vortex`.
4. `NVDLA` (AI Chip with Open NVDLA License):
   `https://github.com/nvdla/hw`.

Based on the parameterized design styles, the projects can be configured with different parameters to generate various hardware. Here, we list the detailed descriptions of the designs in Table A.1.

Table A.1: Descriptions of designs.

| Design | Description |
|---|---|
| RISCY RISCY-FPU zero-riscy | RISCY is a 32-bit CPU core with 4 pipeline stages. RISCY-FPU supports additional RV32F ISA[†] and is configured with FPU[‡]. zero-riscy is a 32-bit CPU with 2 pipeline stages. |
| OpenC910-1 | OpenC910 is a 64-bit CPU core configured with 9∼12 pipelines. |
| Vortex-large Vortex-small | Vortex-large is configured with 8 cores, 4 warps, 8 threads, and L2 Cache. Vortex-small is configured with 1 core, 4 warps, and 4 threads. |
| NVDLA-large NVDLA-small | NVDLA-large is configured with nv_large spec. NVDLA-small is configured with nv_small spec. |

[†] ISA refers to Instruction Set Architecture.
[‡] FPU is Floating Point Unit.

### A.2  DESIGN FLOW

We demonstrate the detailed description of the design flow, from abstract Verilog code (Code.) to the layout after the routing stage, in Figure A.1. Figure A.1a is a module written in abstract Verilog code and Figure A.1b is a synthesized netlist composed of multiple logic cells, e.g., AND, OR, NOT, and Flip-Flop (FF) (Flip-flop.). The logic synthesis stage is conducted between Figure A.1a and Figure A.1b. All the stages from Figure A.1b to Figure A.1g belong to physical design.

Figure A.1c refers to the layout generated by the floorplan stage, where only the positions of two large MACROs are determined. Note that we skip drawing the MACROs in the netlist (Figure A.1b) for simplicity. Besides, the grey lines divide the layout into placement rows such that all the logic cells are aligned in the horizontal direction. Powerplan determines the wires for power supply signals in the layout, as shown in Figure A.1d. Power wires are usually parallel lines on the same metal layer and perpendicular to adjacent metal layers. In this example, horizontal blue lines are used to connect the power supply pins on cells directly; vertical red wider lines are used to deliver power across to the chip layout in a more even distribution. Floorplan and powerplan are two early stages that will decisively influence the subsequent stages and often require some experiential settings and parameters for better optimization. Most settings in our design flow are conducted in these two stages to generate variations of samples.

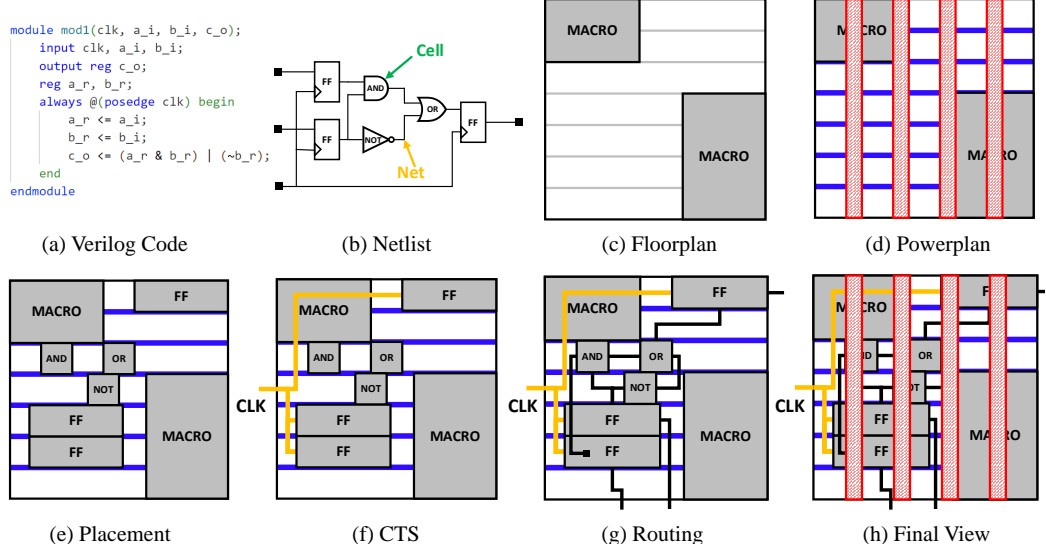

Figure A.1: Detailed visualization of chip design flow, from the abstract code to the results of routing.

Table A.2: Settings of synthesis and physical design.

| Stage | Settings | #Vars | Details |
|---|---|---|---|
| Synthesis | Frequency | 3 | 50/200/500 (MHz) |
| Floorplan | Macro Setting (M)‡ | 4 | - |
| | Utilization (U) | 6 | 50/55/60/65/70/75 (%) |
| | Aspect Ratio (AR) | 3 | 1.0/1.5/2.0 |
| Powerplan | PG Spacing‡ | 8 | - |
| Filler Insertion | Stage Setting | 2 | BR/AR† |

† BR: before routing, AR: after routing.
‡ Macro Setting and PG Spacing refer to different configurations in the EDA tool.

Because the width of vertical power lines will hinder the visualization of subsequent stages, we temporarily hide them in Figure A.1e~g. Figure A.1e shows the layout after placement, where cells are placed in the space that MACROs do not occupy. The positions and directions of cells are decided in this stage. The following stage in Figure A.1 is CTS, which is also known as clock routing (i.e., route clock signals/nets). Note that according to the functionality of cells, only FFs in this example have pins connecting to the clock net. The reason for conducting CTS before routing is that the clock nets are more critical and need more free space to achieve higher design targets. Then, routing is conducted to complete all the functional connections between cells, which is often the most time-consuming stage in the physical design flow. The results of routed nets are drawn in black lines in Figure A.1g. Note that in real scenarios, the routing of a net may go through multiple metal layers to finish the connection. We only draw a simplified version with one metal layer in the example. In Figure A.1h, we combine the previously hidden power lines with the layout after routing. This is the final result of physical design, which can be delivered to foundries for manufacturing.

In Table A.2, we list the parameter variations for the data generation flow to explore the design space and enhance diversity.

## A.3 FEATURES AND LABELS EXPLANATION

In the ML tasks for EDA, the selection of features and labels is critical for the performance of ML models. The source of features includes netlists, layouts, timing data, and power data, which can all

be achieved from CircuitNet. The netlists and layouts provide two ways to encode the geometric, timing, and power data as features in graph mode or image mode. Some features are easy to utilize for their format is similar to raw data, e.g. positions in the layout. However, due to the rigid data format of raw data extracted from EDA flow, additional hand-crafted features generated at a little cost are employed to facilitate the ML models to enhance prediction ability. In this section, we introduce the features and labels according to the encoding format, e.g. image or graph.

To illustrate image features, we first define the concept of **grid**, which is used to measure the resolution of the image used for our tasks. One grid of the layout is the same as one pixel of the image. Therefore, the size of grids will influence the performance of ML models.

- **Image Features**
    1. Macro Region: A 2D array that encodes the chunked regions covered by macros.
    2. Cell Density: A 2D array that encodes the distribution of cell numbers based on the pre-defined grids.
    3. Pin Configuration: A 2D array that encodes the concrete shapes of all pins with high resolution.
    4. RUDY: 2D arrays that encode the routing demand estimation on the grids. It is an abbreviation of *Rectangular Uniform wire DensitY* (Spindler & Johannes, 2007). A variant of it, named pin RUDY (Liu et al., 2021; Xie et al., 2018), is used to encode the pin density estimation on the grids.
    5. eCongestion: 2D arrays that encode the overflow of routing demand on the grids by an extremely rough estimation during placement.
    6. Congestion: 2D arrays that encode the overflow of routing demand on the grids by global routing.
    7. Overall Power: 2D arrays that encode the multiple types of power data on the grids, including internal, switching, and leakage power.
    8. Temporal Power: A 3D array that encodes a fixed-length discrete sampling of 2D time-variant dynamic power map in a clock period.

- **Image Labels**
    1. Congestion: 2D arrays that encode the overflow of routing demand of each grid by global routing.
    2. DRC Violations: A 2D array that encodes the distribution of DRVs number on the grids.
    3. IR-Drop: A 2D array that encodes the distribution of the maximum IR-drop value on the grids.

For graph features, the raw netlist is the core of the graph-based ML model applied for the EDA flow. Based on the topological relationship defined in netlists, multiple properties belonging to cells and nets can be encoded as graph properties in the graph. It should be emphasized that we only demonstrate some utilized features of the ML tasks in the paper.

- **Graph Features**
    1. Graph: A graph extracted from the netlist.
    2. Instance Placement: The graph-encoded properties of the location of all cells and macros extracted from placement results.
    3. Pin Position: The graph-encoded properties of the location of all pins extracted from placement results.

- **Graph Labels**
    1. Net Delay: The graph-encoded properties of the time of signal propagation from one pin to another pin of the same net.

## A.4 OTHER DISCUSSIONS

In this section, we discuss the crucial factors related to dataset generation about chip design.

### A.4.1 SYNTHESIS STAGE

In our work, we collect the designs from the open-source hardware community, which also experiences a period of prosperity in recent years. Synthesis is the first critical stage for designers to implement the chip design. The inputs of the synthesis stage are hardware designs and some timing constraints, which are the two factors to enlarge the diversity in this stage. The timing constraints are usually a user-defined parameter, that may vary from 10MHz to 1,000MHz for most open-source hardware. Therefore, the collection and manipulation of designs have more importance in these two factors.

**Design diversity and quality.** Design diversity and quality are the pillars of a dataset for chip design. The goal of our dataset is to maintain diversity and quality at the same time. Therefore, we choose well-maintained open-source hardware projects, e.g. PULPino (Gautschi et al., 2017), OpenC910 (Chen et al., 2020a), Vortex (Tine et al., 2021), and NVDLA (NVDLA). These designs include CPU, GPU, and AI Chip with abundant diversity in the dimension of architecture. As we know, these designs are adopted by many companies and institutes and even undergo silicon-proven, which increases the confidence of these designs. We think these chosen samples may be closer to realistic industrial hardware designs in terms of diversity and quality.

**Design manipulation.** Design manipulation is a challenging problem we have to face when creating the dataset. For example, even for highly-developed open-source hardware projects, some Intellectual Property (IP) modules are hard to acquire by ordinary users. The Static Random-Access Memory (SRAM) IPs used in the designs are based on artificial modules, which is critical to complete the whole flow. We add these artificial modules directly in the RTL designs without manipulating the outcome netlists of synthesis tools, which better honors the optimization of commercial synthesis tools.

### A.4.2 PHYSICAL DESIGN STAGE

Physical design is the highlight of our dataset compared with other works. The stage of physical design is so time-consuming that samples are hard to get for most researchers. Current benchmarks from the EDA contests provide only isolated data for specific tasks, which is not suitable for cross-stage optimization and ML tasks on these raw geometric format data. These factors restrict the innovations of ML in the field of chip design. To explain the contribution of our work, design space in the physical design stage and suitable data format for ML are discussed here.

**Design Space.** The design space in physical design is also large considering the massive parameters to be set in the design flow. In our work, we focus on the floorplan and powerplan stage, where many parameters need to be set by human experts. Due to these two stages being in the early portion of physical design, many consistent samples will be generated targeted at congestion, DRV, IR_drop, and timing.

**Data Organization.** In the whole flow of physical designs, massive data are generated containing spatial, temporal, and topological information. However, most data are written in document format, which incurs a time-consuming step for the researchers to extract these data in a suitable format to process. In our work, we provide the proper sanitized data, considering the NDA problems, in both raw data and Python data format. This data organization is suitable for researchers from the ML community, who focus on the innovation of ML models, and researchers from the EDA community, who try to hack into the detailed physical design flow.

## B COMPLEMENTARY PREDICTION TASKS

In this section, we further explain the other two ML prediction tasks, e.g., routability prediction and IR-drop prediction. Meanwhile, we performed experiments on CircuitNet 2.0 and CircuitNet.

### B.1 ROUTABILITY PREDICTION

We take congestion prediction as an example to explain the details of routability prediction.

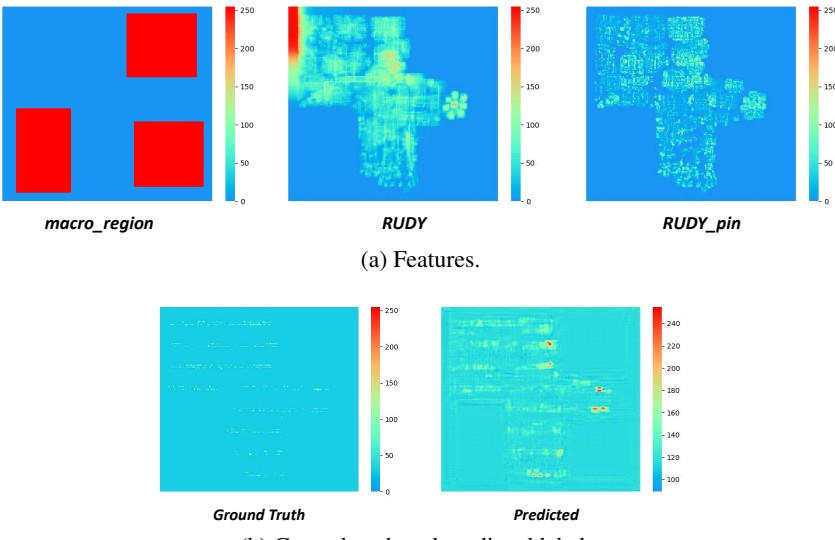

(a) Features.

(b) Ground truth and predicted labels.

Figure B.1: Visualization of the congestion prediction.

### B.1.1 TASK DESCRIPTION

Congestion is a metric to evaluate the density of routed wires on the layout, which is often utilized to optimize the positions of macros and cells in the floorplan and placement stages. As the precious congestion data can only be achieved by time-consuming global routing, the fast prediction model is attractive for accelerating the design flow. Therefore, the goal of congestion prediction models is leveraging the data from the placement stage to predict the congestion map after global routing.

We define labels to predict by the congestion maps, whose formats are usually 2D arrays just like images. The input features can be extracted from the layout definition file and the netlists. For example, the positions of macros and cells, the topological connections of nets, and related density information calculated by the placement results can be designed as features to learn the behavior of global routing.

### B.1.2 EXPERIMENTS

Congestion prediction can be formulated into image-to-image translation, and generative models, like Fully Convolutional Networks (FCN) (Liu et al., 2021) and conditional Generative Adversarial Neural Networks (cGAN) (Alawieh et al., 2020), can be applied in this task to process the geometric information in the layout. Congestion prediction can also be completed with GNN (Yang et al., 2022), to consider not only the geometric information but also the topological information in the design. We use the GNN model (Yang et al., 2022) based on their open-source repository and follow their setup to build a graph on CircuitNet and train the model.

For FCN and cGAN, they have the same input features and label, and we set the batch size to 16 and the learning rate to $2 \times 10^{-4}$. For GNN, we set the batch size to 1 and the learning rate to $2 \times 10^{-4}$. We train the models five times with no random seeds to collect the error bar. We use Adam Optimizer for all experiments.

**Input features**: Encode features in image format: *macro_region*, *RUDY*, and *RUDY_pin* shown in Figure B.1a. Encode features in graph format: A heterogeneous graph of cells and nets as nodes, and topological connections and geometric connections as edges.

**Ground truth**: *Congestion map* shown in Figure B.1b.

**Evaluation metrics:** The method to evaluate the quality of congestion is to measure the similarity between the actual and predicted congestion map. For models with image-based training manner, including FCN and GAN, the label and prediction are image-like arrays, so we follow the previous work to use NRMSE and SSIM to evaluate the pixel-level accuracy and image similarity. For the

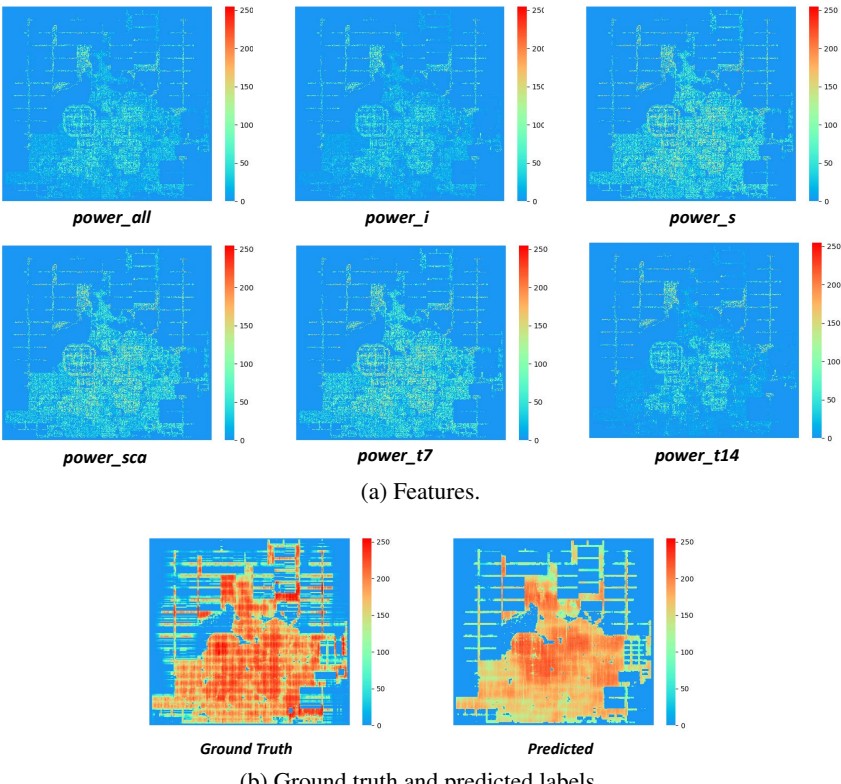

(a) Features.

(b) Ground truth and predicted labels.

Figure B.2: Visualization of IR-drop prediction.

GNN model, its prediction is the node features on the graph, which is more fine-grained. We convert the prediction to a congestion map through tiling to enable comparison and take the maximum value in each tile as the final result.

**Results**: In this experiment, we experiment on both CircuitNet 2.0 and CircuitNet. The results are shown in Table B.1. The first observation is that all models show relatively good performance on both datasets, which indicates our data are properly collected to support existing methods. Moreover, all methods have higher performance on CircuitNet 2.0, which indicates the more diverse training set in CircuitNet 2.0 leads to better generalization on unseen designs in the test set.

Table B.1: Results of congestion prediction in NRMSE, SSIM.

| Dataset | Model | NRMSE | SSIM |
| --- | --- | --- | --- |
| CircuitNet 2.0 | FCN | $0.036 \pm 0.001$ | $0.872 \pm 0.005$ |
| CircuitNet 2.0 | cGAN | $0.044 \pm 0.003$ | $0.796 \pm 0.007$ |
| CircuitNet 2.0 | Circuit GNN | $0.020 \pm 0.001$ | $0.870 \pm 0.009$ |
| CircuitNet | FCN | $0.047 \pm 0.002$ | $0.773 \pm 0.006$ |
| CircuitNet | cGAN | $0.055 \pm 0.003$ | $0.739 \pm 0.008$ |
| CircuitNet | Circuit GNN | $0.046 \pm 0.001$ | $0.761 \pm 0.003$ |

## B.2 IR-DROP PREDICTION

### B.2.1 TASK DESCRIPTION

IR-drop is a metric about the stability and robustness of the Power Delivery Network (PDN), where the lower values mean fewer potential failures of chips. The IR-drop data is a distribution across the layout affected by the power distribution of cells and PDN. The hotspots in the power distribution

Table B.2: Results of IR-drop prediction with error bars.

| Dataset | Model | FPR (<=5%) | TPR (%) | Accuracy (%) | AUC (ROC) |
|---------|-------|------------|---------|--------------|-----------|
| CircuitNet 2.0 | MAVIREC | 4.7 | $39.6 \pm 1.9$ | $95.2 \pm 0.1$ | $0.8671 \pm 0.0032$ |
| CircuitNet | MAVIREC | 4.4 | $29.7 \pm 1.2$ | $95.4 \pm 0.1$ | $0.8537 \pm 0.0023$ |

map may trigger related cells to encounter IR-drop degradations, where the heavy IR-drop regions are more critical for the designs. In the design flow, the power distribution of cells is utilized to analyze the IR-drop in the layouts. The goal of IR-drop prediction is to utilize ML models to substitute traditional analytic methods at a faster speed, in order to detect the hotspots of IR-drop distribution maps.

In the formulation of IR-drop prediction, IR-drop hotspot maps are the labels to be predicted. The input features are different types of power maps including static and dynamic power data. Different from 2D static data, dynamic data is a 3D array encoding both spatial and temporal information. Two dimensions of the dynamic power map represent the spatial distribution across the layout. Because the transient dynamic power data is temporally continuous within a clock period, the pre-defined time interval and probability of signal switching are used to sample the dynamic data in a discrete format.

### B.2.2 EXPERIMENTS

Similar to the DRV prediction, IR-drop prediction is also formulated into a hotspot detection task, and CNN models, like FCN, can be applied in this task to first predict with image-to-image translation, then covert the prediction into binary through applying a threshold. We employ MAVIREC (Chhabria et al., 2021c) for our experiments. We use Adam optimizer with a learning rate of $2 \times 10^{-4}$ and L1 Loss for all experiments. We train the models five times with no random seeds to collect the error bar.

**Input features**: *power_all*, total power. *power_i*, internal power. *power_s*, switching power. *power_sca*, toggle rate scaled power. *power_t7*, time-decomposed power in the 7th timing window. *power_t14*, time-decomposed power in the 14th timing window shown in Figure B.2a.

**Ground truth**: *IR-drop* shown in Figure B.2b.

**Evaluation metrics:** The IR-drop hotspots are also much less than non-hotspots, so we also adopt Receiver Operating Characteristic (ROC) curve to evaluate the imbalance classification task.

**Results**: The IR-drop prediction results are shwon in Table B.2. Similar to the results in congestion prediction, the model has higher performance on CircuitNet 2.0, which indicates the more diverse training set in CircuitNet 2.0 leads to better generalization on unseen designs in the test set.

