# OpenReview forum: "CircuitNet 2.0: An Advanced Dataset for Promoting Machine Learning Innovations in Realistic Chip Design Environment"
_ICLR.cc/2024/Conference — ICLR 2024 poster_

### Official Review · Reviewer_StPZ · 2023-10-26

**Soundness:** 2 fair
**Presentation:** 3 good
**Contribution:** 2 fair
**Rating:** 6
**Confidence:** 5

**Summary:**

This paper introduces a new dataset, CircuitNet 2.0, for prediction tasks in the EDA area. The dataset extends CircuitNet with new circuits, labels, and features. The dataset has been verified as effective by some realistic ML tasks.

**Strengths:**

1. Nowadays, the dataset for ML on EDA is still very limited. This dataset can prompt research in this area.

2. The dataset files are very complete, which can help researchers to use them.

3. The paper also includes the analysis of the dataset, exploring some possible usage of it, such as the timing analysis. Also, it discusses some problems and solutions, such as the imbalance of data.

4. The data acquisition is expensive. According to the paper, some designs cost over one week. The work can help researchers save their time and money.

**Weaknesses:**

1. The dataset can be seen as an extension of CircuitNet with a few new features and labels, which can be easily obtained from the synopsis tools. For the number of designs, CircuitNet 1.0 and 2.0 both contain about 10,000 designs. According to experience, doubling the amount of data has limited improvement in prediction accuracy.

2. Prediction tasks using the dataset are not the final task for chip design tasks. In other words, the most significant tasks in EDA are decision tasks. Thus, the impact of this data set still appears to be limited.

**Questions:**

Will the trained parameters of the timing, routing, and IR-drop prediction models be open-sourced to reproduce the results of Table 3, B1, and B2?

---

> ### Author Response · Authors · 2023-11-19
> **Reply to comments.**
>
> Thanks very much for your valuable feedback on our paper. We carefully read the comments, improved our paper, and provided detailed replies to your question as follows.
>
> > Q1: "The dataset can be seen as an extension of CircuitNet with a few new features and labels, which can be easily obtained from the synopsis tools. For the number of designs, CircuitNet 1.0 and 2.0 both contain about 10,000 designs. According to experience, doubling the amount of data has limited improvement in prediction accuracy."
>
> CircuitNet 2.0 is not an extension of CircuitNet 1.0 by just increasing the number of features and labels.
> Firstly, including GPU and AI chips in CircuitNet 2.0 enables new machine learning tasks, as described in Section 4.2, compared with merely CPU designs in CircuitNet 1.0. These tasks have more realistic meanings, which probably ignites researchers to deeply think about how to devise effective ML algorithms in a realistic design environment.
> Secondly, the advanced 14nm commercial technology is harder to access even compared with 28nm technology used by CircuitNet 1.0. The open-source commercial PDKs are still 180mm [1] and 130nm [2].  The academic predictive PDK [3] lacks reliability and precision compared with commercial PDKs.
>
> Therefore, CircuitNet 2.0 is the most advanced IC design dataset,  which largely improves the foundation of ML for EDA research.
>
> [1] https://gf180mcu-pdk.readthedocs.io/en/latest/
>
> [2] https://skywater-pdk.readthedocs.io/en/main/
>
> [3] https://github.com/The-OpenROAD-Project/asap7
>
> > Q2: "Prediction tasks using the dataset are not the final task for chip design tasks. In other words, the most significant tasks in EDA are decision tasks. Thus, the impact of this data set still appears to be limited."
>
> The successful decision results in EDA are based on accurate estimation of target metrics, which is the common pursuit of classic methods and ML-based methods. CircuiNet 2.0 provides almost all data for decision tasks in EDA, including layout, netlist, power, and timing. For example, our dataset fully supports these recent works [1-5], which integrate ML models to help the decisions of classic optimization methods. In a word, CircuitNet 2.0 will help the community explore the boundary of decision tasks in EDA by combining classic methods and ML-based methods.
>
> [1] Jiang et al. Accelerating Routability and Timing Optimization with Open-Source AI4EDA Dataset CircuitNet and Heterogeneous Platforms, ICCAD'23
>
> [2] Liu et al., “Concurrent Sign-off Timing Optimization via Deep Steiner Points Refinement”, ACM/IEEE Design Automation Conference (DAC), San Francisco, Jul. 09–13, 2023.
>
> [3] Zheng et al., “Mitigating Distribution Shift for Congestion Optimization in Global Placement”, ACM/IEEE Design Automation Conference (DAC), San Francisco, Jul. 09–13, 2023.
>
> [4] Liu et al., “Global Placement with Deep Learning-Enabled Explicit Routability Optimization”, IEEE/ACM Proceedings Design, Automation and Test in Europe (DATE), Feb. 01–05, 2021.
>
> [5] Mirhoseini et al., "A graph placement methodology for fast chip design", Nature, 594(7862) (2021), pp. 207-212.
>
> > Q3: "Will the trained parameters of the timing, routing, and IR-drop prediction models be open-sourced to reproduce the results of Table 3, B1, and B2?"
>
> Yes, we will open-source the trained parameters for the reproduction of our experiments.

---

### Official Review · Reviewer_Xf2K · 2023-10-28

**Soundness:** 3 good
**Presentation:** 3 good
**Contribution:** 3 good
**Rating:** 5
**Confidence:** 4

**Summary:**

This paper introduces an advanced large-scale dataset for chip design called CircuitNet 2.0 which aims to motivate the ML development for EDA. Specially, over 10,000 samples including CPU, GPU, and AI Chip are collected and multi-modal features with labels in the dataset can be used in various prediction tasks, such as routability, timing, and power. Experiments on multiple realistic machine learning tasks, i.e., learning on imbalanced data and transfer learning are introduced to verify its potential.

**Strengths:**

1. The development of a large-scale public dataset is crucial for the advancement of ML algorithms in EDA area. This work provides rich design types for practical prediction tasks.
2. The experiments on imbalanced data and transfer learning are reasonable and sound.

**Weaknesses:**

1. The large-scale dataset which contains more than 10000 samples is originated from only eight open-source designs, especially three smaller CPU designs. It is likely that samples from the same design share similar features.
2. Note that most settings in the design flow are conducted in the floorplan and powerplan stages, some combination of parameters may be unreasonable compared to those realistic industrial designs.

**Questions:**

1. The t-SNE plot in section 4.2.2 is confusing. Where are the GPU and AI Chip?
2. In the transfer learning experiment, how long does it take to train 20000 iterations?
3. Could the authors verify the advantage of training with multi-modal data? It seems that only graph-based input is adpoted in section 4.1.

---

> ### Author Response · Authors · 2023-11-19
> **Reply to comments.**
>
> Thanks very much for your valuable feedback on our paper. We carefully read the comments, improved our paper, and provided detailed replies to your question as follows.
>
> > Q1: "The large-scale dataset which contains more than 10000 samples is originated from only eight open-source designs, especially three smaller CPU designs. It is likely that samples from the same design share similar features."
>
> CPU is the most typical hardware in the realistic design environment. We list the short descriptions of these CPU designs in Table A.1, which shows the variations among CPU-type designs are still significant. Besides, we demonstrate the t-SNE results of the features of CPU designs in Figure 5, which shows a reasonable distribution for ML tasks.
>
> > Q2: "Note that most settings in the design flow are conducted in the floorplan and powerplan stages, some combination of parameters may be unreasonable compared to those realistic industrial designs."
>
> These parameter settings have been extensively verified by several experts in IC design who have been working for leading IC companies. We believe this dataset can provide a valuable foundation for ML for EDA research.
>
> > Q3: "The t-SNE plot in section 4.2.2 is confusing. Where are the GPU and AI Chip?"
>
> We find the missing points of GPU and AI Chip are caused by the incompatible vector format of Figure 5 under different PDF viewers. We have changed the format of Figure 5 from vector to PNG.
>
> > Q4: "In the transfer learning experiment, how long does it take to train 20000 iterations?"
>
> It takes 148 mins to train 20000 iterations with 1 A800 GPU.
>
> > Q5: "Could the authors verify the advantage of training with multi-modal data? It seems that only graph-based input is adpoted in section 4.1."
>
> Thank you for the comment. Recently, Wang et al. [1] have shown that fusion of multi-modal data can outperform purely graph-based or image-based methods. Our dataset provides all the data required to build such a model. As their model has not been open-source yet, we do not have enough time to conduct the experiments during the rebuttal period. We will reproduce their experiments on our dataset and release the sample code in the future.
> Besides, HybridNet [2] and Lay-Net [3] are multi-modal neural networks for congestion prediction, which also indicates the advantages of multi-modal learning in the EDA field.
>
> [1] Wang et al. "Restructure-Tolerant Timing Prediction via Multimodal Fusion." 2023 60th ACM/IEEE Design Automation Conference (DAC). IEEE, 2023.
>
> [2] Zhao et al. "HybridNet: Dual-Branch Fusion of Geometrical and Topological Views for VLSI Congestion Prediction" arXiv:2305.05374
>
> [3] Zheng et al. “Lay-Net: Grafting Netlist Knowledge on Layout-Based Congestion Prediction”, International Conference on Computer-Aided Design (ICCAD), 2023

---

### Official Review · Reviewer_uzNQ · 2023-11-05

**Soundness:** 3 good
**Presentation:** 3 good
**Contribution:** 2 fair
**Rating:** 6
**Confidence:** 4

**Summary:**

This paper introduces CircuitNet2.0, an open-source EDA dataset, containing data to support multi-model prediction tasks in order to promote ML innovations in a realistic chip design.

**Strengths:**

1. Compared to previous efforts, CircuitNet 2.0 is a more advanced large-scale dataset: 1) provides data for large-scale CPU, GPU, and AI chips with millions of gates; 2) is based on advanced 14nm FinFET technologies; and 3) supports 3 kinds of tasks, including routability, IR-drop, and timing.
2. This paper explains the dataset (e.g., feature, label) generation flow along with the EDA tool flow, which can inspire the following works to contribute to the ML EDA dataset.
3. This paper demonstrates two realistic ML tasks using CircuitNet2.0 to show the how to solve the potential challenges in ML EDA datasets.

**Weaknesses:**

1. The paper mentions that the current most efficient method for EDA problems is combining ML model with the classical heuristic optimization methods: using ML models to predict tool outcomes at early design stages and guide heuristic optimization has become a popular approach in EDA. In the evaluation part, only the pure ML prediction outputs are evaluated, the overall performance combining ML model with the classical heuristic optimization methods is not shown. It would be better if the authors could show such an overall performance. Since training ML models are also costly, there may be no need to introduce ML models to every P&R stage.

**Questions:**

1. If possible, can the author show several examples to demonstrate the benefit of the overall performance (e.g., P&R time) by combining ML model with the classical heuristic optimization methods?
2. Considering this paper targets EDA problems, can conferences such as DAC and ICCAD be a more suitable platform for publishing this paper?

---

> ### Author Response · Authors · 2023-11-19
> **Reply to comments.**
>
> Thanks very much for your valuable feedback on our paper. We carefully read the comments, improved our paper, and provided detailed replies to your question as follows.
>
> > Q1: "The paper mentions that the current most efficient method for EDA problems ... no need to introduce ML models to every P&R stage"
>
> The combination of ML models and classical optimization methods is inevitable in order to achieving better outcomes in EDA design stages. Integrating ML prediction models into the classical heuristic optimization methods to help decision making has been extensively explored in the EDA community [1-4].
>
> [1] Jiang et al. Accelerating Routability and Timing Optimization with Open-Source AI4EDA Dataset CircuitNet and Heterogeneous Platforms, ICCAD'23
>
> [2] Liu et al., “Concurrent Sign-off Timing Optimization via Deep Steiner Points Refinement”, ACM/IEEE Design Automation Conference (DAC), San Francisco, Jul. 09–13, 2023.
>
> [3] Zheng et al., “Mitigating Distribution Shift for Congestion Optimization in Global Placement”, ACM/IEEE Design Automation Conference (DAC), San Francisco, Jul. 09–13, 2023.
>
> [4] Liu et al., “Global Placement with Deep Learning-Enabled Explicit Routability Optimization”, IEEE/ACM Proceedings Design, Automation and Test in Europe (DATE), Feb. 01–05, 2021.
>
> > Q2: " If possible, can the author show several examples to demonstrate the benefit of the overall performance (e.g., P&R time) by combining ML model with the classical heuristic optimization methods?"
>
> According to references in answers to Q1, Jiang et al. [1] integrate ML model with classical optimization methods to improve routability and timing optimization based on CircuitNet 1.0. Liu et al. [2] use GNN model to optimize the Steiner points of routing solution. Zheng et al. [3] and Liu et al. [4] use ML models to predict the routability for the optimization of placement.
>
> > Q3: "Considering this paper targets EDA problems, can conferences such as DAC and ICCAD be a more suitable platform for publishing this paper?"
>
> ML for EDA is already a hot topic in DAC/ICCAD community and our dataset has attracted some attention. As many problems in EDA field correspond to fundamental graph and geometric problems, which are very different from current main ML applications like computer vision, social media, natural language processing, etc., and cannot be easily solved by existing ML techniques. Thus, we would like to promote this CircuitNet 2.0 dataset to a broader ML community to bring EDA problems to the attention of ML researchers, stimulate fundamental breakthroughs in ML for EDA techniques, and broaden ML applications. Therefore, we submit our dataset to the well-known ML conference, ICLR.
>
> To achieve these goals, CircuitNet 2.0 provides user-friendly feature data generated by EDA tools. Considering the hardness of accessing EDA data, we think public realistic EDA data is valuable for the ML researchers and will build a bridge between ML community and EDA community.

---

### Official Review · Reviewer_XfNZ · 2023-11-07

**Soundness:** 3 good
**Presentation:** 3 good
**Contribution:** 4 excellent
**Rating:** 6
**Confidence:** 4

**Summary:**

The main contribution of this work is to propose a dataset called CircuitNet 2.0. Due to the release prohibition and human overload, the research field of EDA lacks sufficient datasets. Even those existing datasets are mostly small. Huge and high-quality datasets are valuable, so the proposition of such datasets basically has value. Compared with the previous large dataset CircuitNet, this 2.0 version is more capable in design types, advanced technologies, and timing prediction tasks. The dataset has a total of 10,791 samples, which is similar to CircuitNet.

**Strengths:**

+ Compared with CircuitNet which only considers CPU, CircuitNet 2.0 has more design types, including CPU/GPU/AI chip, which enriches the diversity of chip designs and benefits the approximation to the distribution of chip designs.
+ The samples in CircuitNet 2.0 are absolutely different from the ones in CircuitNet, so these two datasets could be used simultaneously and might train a better model.
+ CircuitNet 2.0 uses more advanced 14nm FinFET technologies and displays the additional timing prediction task that CircuitNet does not perform.

**Weaknesses:**

- Each sample in the same design has the same number of cells, nets, macros, pins, and IOs, so this dataset is not suitable, to some extent, for the majority of ML approaches since the training would easily lead to overfitting. The authors lack the discussion with regard to overfitting and latent negative impacts.
- This is not a severe weakness but the author claims in Section 4.1.2 that the performance of MLP is lower than that of GNN, which leaves a doubt because the GNN is delicately devised but the devise of MLP seems to be arbitrary.
- Please refer to some other concerns and questions in `Questions’.
- typo: line 12th in section 4.3, two `introduce’.

**Questions:**

- I note that the samples took a very long time to generate. What is the total generation time? How many and what kind of machines are used to generate these samples?
- What about the results of congestion prediction on the training set? For example, what about the NRMSE and SSIM on CPU cases when using CPUs as training data?
- What does `details’ mean in Table A.2? What are the numbers below?
- Is the learning rate the same in transfer learning? (Figure 6)
- In the experiments of congestion prediction and IR-drop prediction, models using CircuitNet 2.0 as training data have more advanced performance than those using CircuitNet. In these two experiments, what are the `unseen designs in the test set’?

---

> ### Author Response · Authors · 2023-11-19
> **Reply to comments.**
>
> Thanks very much for your valuable feedback on our paper. We carefully read the comments, improved our paper, and provided detailed replies to your question as follows.
>
> > Q1: "Each sample in the same design ... latent negative impacts."
>
> The statistics listed in Table 1 are the average numbers of components in netlists after the synthesis stage in order to show the diversity among different hardware designs. All samples even originated from the same design will contain different numbers of components and nets after the synthesis and physical design stages due to various optimization steps to restructure the netlists. Note that even the physical design stage will change the circuit netlists.
>
> To address the concern of overfitting, we perform K-fold experiments in Table 4 to show the generalization capability between samples of different designs. We can see that the model can still behave in a preferable generalization manner between different designs, which indicates the overfitting and latent negative impacts are limited.
>
> > Q2: "This is not a severe weakness ... MLP seems to be arbitrary."
>
> We compare the performance of MLP and GNN to show the importance of modeling topological relationships within netlists. Barboza et al. [1] propose an MLP for timing prediction and Guo et al. [2] use their MLP as a baseline. Our MLP is a well-tuned model similar to Guo et al. [2].
>
> [1] Barboza et al., "Machine LearningBased Pre-Routing Timing Prediction with Reduced Pessimism.", ACM/IEEE Design Automation Conference (DAC), 2019.
>
> [2] Guo et al. "A Timing Engine Inspired Graph Neural Network Model for Pre-Routing Slack Prediction", ACM/IEEE Design Automation Conference (DAC), 2022.
>
> > Q3: "typo: line 12th in section 4.3, two `introduce’."
>
> We have fixed it in the revision.
>
> > Q4: "I note that the samples took a very long time to generate. What is the total generation time? How many and what kind of machines are used to generate these samples?"
>
> Our generation flow consists of data collection, SRAM generation, verilog code integration, logic synthesis and physical design using commercial tools, manual data check, feature extraction, feature check and model-based verification. We take about 1 year to generate and verify these samples. We use one 96-core Linux server with Intel(R) Xeon(R) Gold 6248R CPU @ 3.00GHz and 512 GB memory to generate these samples.
>
> > Q5: "What about the results of congestion prediction on the training set? For example, what about the NRMSE and SSIM on CPU cases when using CPUs as training data?"
>
> We divide CPU designs into two sets, i.e., zero-riscy + OpenC910-1 and RISCY+RISCY-FPU, for congestion prediction as an additional experiment. The results are shown as follows. The average NRMSE is 0.0202 and SSIM is 0.8984 in the additional experiment.
>
> When we use CPU as the training set, the prediction results on GPU and AI chip in Table 4 are worse than those on CPU. The comparison indicates the similarity within same-type designs really exists and also further justifies the value of GPU and AI Chip to increase the diversity of our dataset.
>
> Use zero-riscy + OpenC910-1 as Train Set.
> |Test Set|NRMSE|SSIM|
> |--|--|--|
> |RISCY|0.0203|0.8989|
> |RISCY-FPU|0.0214|0.7195|
>
> Use RISCY + RISCY-FPU as Train Set.
> |Test Set|NRMSE|SSIM|
> |--|--|--|
> |zero-riscy|0.0184|0.9183|
> |OpenC910-1|0.0366|0.7195|
>
> > Q6: "What does `details’ mean in Table A.2? What are the numbers below?"
>
> The ‘details’ means the specific values for the settings of commercial tools. The numbers below are exactly the values for different settings. For 'Macro Setting' and 'PG Setting', different commands are used in tcl scripts, so we fill in '-'. We add '(MHz)' to the details of 'Frequency' and '(%)' to the details of 'Utilization (U)' to clarify the explanations.
>
> > Q7: "Is the learning rate the same in transfer learning? (Figure 6)"
>
> Yes, the learning rate in transfer learning is the same as that in training from scratch, which is 2e-4, because we use Adam optimizer which has adaptive learning rate, and we find the initial learning rate setting doesn't affect the training results. We also add the clarification in Section 4.3.
>
> > Q8: "In the experiments of congestion prediction and IR-drop prediction, models using CircuitNet 2.0 as training data have more advanced performance than those using CircuitNet. In these two experiments, what are the `unseen designs in the test set’?"
>
> The unseen designs are chosen from "ISPD 2015 Benchmarks with
> Fence Regions and Routing Blockages for Detailed-Routing-Driven Placement." (https://ispd.cc/contests/15/web/benchmark.html). These designs are provided by the contest organizers and are completely different from the samples included in the CircuitNet 2.0.

---

> > ### Comment · Reviewer_XfNZ · 2023-11-22
> > **Follow-up to author response**
> >
> > Thanks for your response to my review and questions. I will take it into consideration for my final score.

---

> > ### Comment · Reviewer_XfNZ · 2023-12-03
> >
> > I appreciate the authors’ response and acknowledge that they have addressed the majority of my questions and concerns. My remaining concern pertains to the diversity of the dataset, which is derived from a limited sample of 8 cases. This issue was also highlighted by reviewer `Xf2K`. Although the authors have demonstrated the model’s generalization capabilities, the potential for latent overfitting still exists. However, given the prevalence of closed-source datasets in the EDA community, it might be overly stringent to demand the authors to address the inherent diversity. Therefore, I have decided to increase my final score to 6.

---

### Meta-Review · Area_Chair_acQa · 2023-12-06

**Metareview:**

This work introduces a huge and high-quality dataset named CircuitNet 2.0 for EDA research, containing information on multiple design types, advanced technologies, and timing prediction tasks. The reviewers show that such a dataset can inspire more ML works for EDA. Nevertheless, the reviewers also point out the limitations, e.g., the information is collected from only eight open-source designs, which can lead to overfitting for ML methods; Also, it is better to provide the overall performance combining ML model with the classical heuristic optimization methods. Taking these mixed opinions into account, I lean toward acceptance.

**Justification For Why Not Higher Score:**

Please see the meta-review

**Justification For Why Not Lower Score:**

Please see the meta-review

---

### Decision · Program_Chairs · 2024-01-16

Accept (poster)